# L444P *Gba1* mutation increases formation and spread of α-synuclein deposits in mice injected with mouse α-synuclein pre-formed fibrils

Anna Migdalska-Richards[1,2☯]*, Michal Wegrzynowicz[3,4☯], Ian F. Harrison[5], Guglielmo Verona[6], Vittorio Bellotti[6], Maria Grazia Spillantini[3], Anthony H. V. Schapira[1]*

**1** Department of Clinical Neurosciences, Institute of Neurology, University College London, London, United Kingdom, **2** Complex Disease Epigenetics Group, University of Exeter, Royal Devon & Exeter Hospital, Exeter, United Kingdom, **3** Department of Clinical Neurosciences, University of Cambridge, Cambridge, United Kingdom, **4** Laboratory of Molecular Basis of Neurodegeneration, Mossakowski Medical Research Centre Polish Academy of Sciences, Warsaw, Poland, **5** Centre for Advanced Biomedical Imaging, University College London, London, United Kingdom, **6** Centre for Amyloidosis and Acute Phase Proteins, Faculty of Medical Sciences, University College London, London, United Kingdom

☯ These authors contributed equally to this work.
* a.schapira@ucl.ac.uk (AHVS); a.migdalska-richards@exeter.ac.uk (AMR)

## Abstract

Parkinson disease is the most common neurodegenerative movement disorder, estimated to affect one in twenty-five individuals over the age of 80. Mutations in glucocerebrosidase 1 (*GBA1*) represent the most common genetic risk factor for Parkinson disease. The link between *GBA1* mutations and α-synuclein accumulation, a hallmark of Parkinson disease, is not fully understood. Following our recent finding that *Gba1* mutations lead to increased α-synuclein accumulation in mice, we have studied the effects of a single injection of mouse α-synuclein pre-formed fibrils into the striatum of *Gba1* mice that carry a L444P knock-in mutation. We found significantly greater formation and spread of α-synuclein inclusions in *Gba1*-transgenic mice compared to wild-type controls. This indicates that the *Gba1* L444P mutation accelerates α-synuclein pathology and spread.

## Introduction

Parkinson disease (PD) is the most common neurodegenerative movement disorder, estimated to affect 4% of individuals over 80 years of age [1]. The most common risk factor for PD are mutations in the glucocerebrosidase 1 (*GBA1*) gene, which encodes an enzyme (GCase) that is involved in glycolipid metabolism. It has been estimated that at least 7–10% of non-Ashkenazi PD individuals have a *GBA1* mutation (PD-*GBA1*) [2]. Although the molecular mechanisms by which *GBA1* mutations increase PD risk are still unclear, it is likely that α-synuclein accumulation plays an important role.

The link between GCase deficiency, α-synuclein accumulation and neurodegeneration in the substantia nigra has recently been explored in a heterozygous *Gba1* mouse model carrying

**Data Availability Statement:** All relevant data are within the manuscript and its Supporting Information files.

**Funding:** This work was supported by the Parkinson's UK grants G-1403 and G-1704, and Medical Research Council (MRC) grants MR/M006646/1 and MR/N028651/1. A.H.V.S. is supported by the NIHR University College London Hospitals Biomedical Research Centre.

**Competing interests:** The authors have declared that no competing interests exist.

a L444P knock-in mutation (*L444P/+* mice) [3]. A significant decrease in GCase activity was associated with increased α-synuclein accumulation, but with no other PD pathology [3]. Intriguingly, overexpression of human α-synuclein in the substantia nigra resulted in significantly greater loss of nigral dopaminergic neurons in *L444P/+* mice than in their wild-type littermates [3]. These results indicate that the *Gba1* L444P mutation alone is not sufficient to induce overall PD pathology but requires an additional factor such as overexpression of α-synuclein. This may contribute to the partial penetrance of *GBA1* mutations causing PD; it is estimated that only 30% of individuals with *GBA1* mutations will develop PD by the age of 80 [2].

The mechanism of accumulation of misfolded fibrillar α-synuclein into inclusions (known as Lewy bodies (LB) and Lewy neurites (LN)) is not completely clear [4, 5]. Emerging evidence shows that misfolded fibrillar α-synuclein is capable of self-propagation and spreading (leading to subsequent accumulation) between interconnected regions of the brain, suggesting that cell-to-cell transmission of pathological forms of α-synuclein plays a crucial role in PD pathogenesis [4]. The presence of even low levels of aggregated or fibrillar α-synuclein (seeds) greatly enhances α-synuclein polymerization into amyloid fibrils [5]. Synthetic α-synuclein pre-formed fibrils (αSYN-PFFs), i.e. laboratory-generated seeds, have been shown to initiate fibrillization and aggregation of soluble endogenous α-synuclein in primary neuronal cultures derived from wild-type mice [6]. More importantly, a single intracerebral injection of αSYN-PFFs greatly accelerates the onset of neuropathological symptoms in transgenic mice expressing the human α-synuclein A53T mutation [5]. Further, a single intrastriatal injection of αSYN-PFFs is capable of initiating α-synuclein spreading and accumulation in wild-type mice, leading to the development of PD-like α-synuclein pathology in the anatomically-interconnected brain regions, further confirming the contribution of cell-to-cell transmission in α-synuclein pathology [4].

The objective of this study was to analyze the effect of mouse-αSYN-PFF injection into the striatum of *L444P/+* mice. Four months post-injection, we observed significantly increased formation and spread of α-synuclein deposits in *L444P/+* mice compared to their wild-type littermates, indicating that the L444P mutation enhances aggregation of endogenous α-synuclein into pathological deposits.

## Materials and methods

### Mice

B6;129S4-Gbatm1Rlp/Mmnc (000117-UNC) mice expressing a heterozygous knock-in L444P mutation in the murine *Gba1* gene (*L444P/+* mice) were compared to their wild-type littermates [3, 7]. Only male animals were used in the study. Mice were treated in accordance with local ethical committee guidelines and the UK Animals (Scientific Procedures) Act 1986. All procedures were carried out in accordance with Home Office guidelines (UK) and in compliance with the ARRIVE guidelines. Breeding, maintenance and all the experimental procedures concerning both *L444P/+* mice and their wild-type littermates were covered by the project licence 70/7685 issued by the United Kingdom Home Office. This study was approved by the Animal Welfare and Ethical Review Body, University College London.

### Injection material and stereotaxic injections

Purification of recombinant mouse α-synuclein and *in vitro* fibril assembly were performed as previously described [4]. Three-month old mice were anesthetized with isofluorane inhalation and stereotactically injected in the right dorsal striatum (co-ordinates: +0.2mm relative to bregma, +2.0mm from midline, +2.6mm beneath the dura) with 2.5μl of either αSYN-PFFs

(5μg), α-synuclein monomers (5μg) or sterile PBS, as previously described [4]. Four *L444P/+* mice and ten wild-type littermates were injected with αSYN-PFFs, four *L444P/+* mice and five wild-type littermates were injected with α-synuclein monomers, and four *L444P/+* mice and five wild-type littermates were injected with sterile PBS.

## Immunohistochemistry

Mice were killed by $CO_2$ inhalation four months post αSYN-PFF injection, brains were extracted, post-fixed in 4% paraformaldehyde in PBS at 4˚C for one week, then cryoprotected and stored in 30% sucrose (Sigma-Aldrich) in PBS supplemented with 0.1% $NaN_3$ (Sigma-Aldrich) at 4˚C. Coronal brain sections (30μm) were cut using a freezing sledge microtome (Bright). Free-floating section immunohistochemistry was performed as previously described [3], but with the following modifications. 1. Antigen retrieval was achieved by incubating the sections in 70% formic acid at room temperature for 20 minutes. 2. Sections were incubated for 72 hours at 4˚C with rabbit primary antibody specific to α-synuclein phosphorylated at Ser129 (p-αSYN) (Abcam, ab59264) diluted at 1:2000 in PBST. 3. Sections were washed in PBS and mounted on SuperFrost® Plus microscope slides (Thermo Scientific) after staining was developed.

## Experimental design and statistical analyses

To measure the extent of α-synuclein pathology, the number of p-αSYN-positive deposits were counted in two brain regions (striatum and cingulate/motor cortex) in *L444P/+* mice (n = 4) and wild-type littermates (n = 10). Counting was performed in the hemisphere ipsilateral to the injection site at three different coronal planes per animal (AP +1.4, +0.1, -0.5mm from the bregma). A series of counting probes (40x40x10μm) arranged in a two-dimensional array spaced at 300μm intervals were superimposed on the analyzed brain region, and p-αSYN-positive LB- and LN-like structures contained within the counting probe were counted under a 100x objective on an Olympus BX53 microscope. To minimize the effect of subjective bias when assessing the results of αSYN-PFF injection, the counting of p-αSYN-positive deposits was assessor-blind. The number of p-αSYN deposits was normalized to the volume of the probe and an average of all probe sites was calculated for each animal as a biological replicate for statistical purposes. The Student *t*-test was used to compare p-αSYN species densities between *L444P/+* and wild-type mice.

## Results

The distribution of p-αSYN deposits (considered here as markers of synucleinopathy) was analyzed throughout the brains of wild-type and *L444P/+* mice four months post αSYN-PFF injection.

Widespread p-αSYN pathology was observed in the brains of αSYN-PFF-injected wild-type mice in the form of LB- and LN-like structures. In the hemisphere ipsilateral to the injection, the highest concentration of p-αSYN-positive inclusions was found in the cortex (especially the parietal, insular, perirhinal and entorhinal cortices and layer 5 of the motor cortex) and in the amygdala (Figs 1A and 1B and S1A and S1B). Prominent accumulation of p-αSYN was also observed in the striatum, layer 2 of the motor cortex, layer 5 of the cingulate cortex and in the substantia nigra pars compacta (SNpc) (Figs 1C, 2A–2C and S1C and S2A–S2C). Although similar regions were affected in the hemisphere contralateral to the injection site, pathology in most analyzed regions was less prominent than in the equivalent ipsilateral region, and was completely absent in the SNpc. Within the motor cortex, pathology in the contralateral

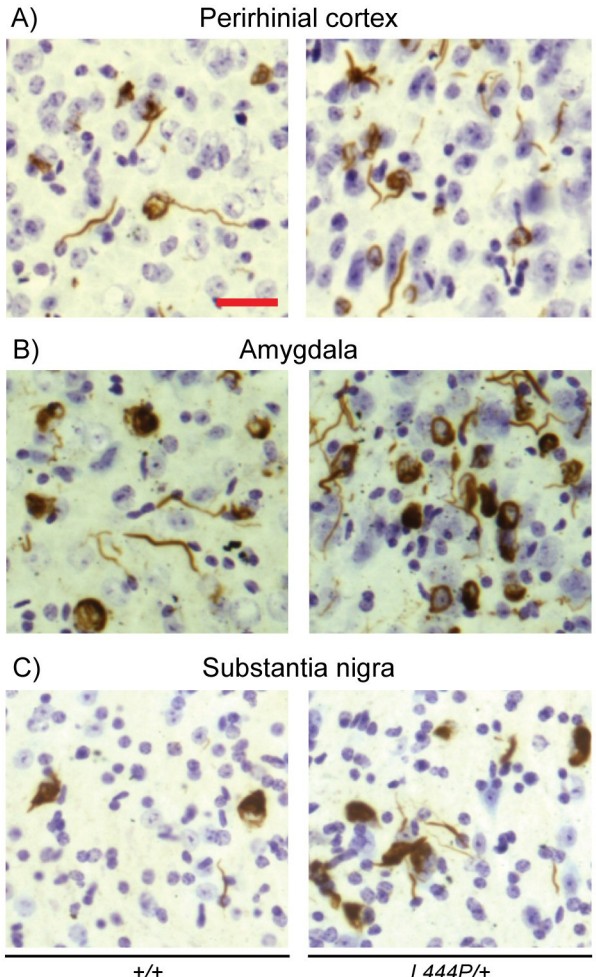

**Fig 1. p-αSYN inclusions in the perirhinal cortex, amygdala and substantia nigra in αSYN-PFF-injected wild-type and *L444P/+* mice.** (**A**) Increased p-αSYN pathology in the perirhinal cortex at the level of 2.2mm posterior to the injection site (-2.0mm from the bregma) in the ipsilateral hemisphere of *L444P/+* mice compared to their wild-type control littermates. (**B**) Increased p-αSYN pathology in the lateral amygdaloid nuclei at the level of 2.2mm posterior to the injection site (-2.0mm from the bregma) in the ipsilateral hemisphere of *L444P/+* mice compared to their wild-type control littermates. (**C**) Increased p-αSYN pathology in substantia nigra pars compacta at the level of 3.7mm posterior to the injection site (-3.5mm from the bregma) in the ipsilateral hemisphere of *L444P/+* mice compared to their wild-type control littermates. Scale bars = 25μm. Representative images shown. In total ten +/+ and four *L444P/+* mice were analyzed.

hemisphere was more prominent in layer 2 of the motor cortex in relation to layer 5 than in the ipsilateral hemisphere (Figs 2A–2C and S2A–S2C).

αSYN-PFF inoculation of *L444P/+* mice resulted in p-αSYN pathology in the same brain regions as in wild-type littermates. The density of p-αSYN deposits, however, appeared to be higher than in wild-type mice in several regions, including the motor, cingulate, perirhinal and entorhinal cortices, amygdala and SNpc (Figs 1A–1C, 2B and 2C and S1A–S1C, S2B and S2C). The extent of p-αSYN pathology was estimated in the ipsilateral hemisphere in the striatum and in the cortical area including the cingulate and motor cortex by counting LB- and LN-like deposits in αSYN-PFF-injected *L444P/+* and wild-type mice. There was a trend for an increase in p-αSYN deposits in the striatum of *L444P/+* mice compared to wild-type littermates, but this was not statistically significant (Student *t*-test, p = 0.23) (Fig 2D). However, a

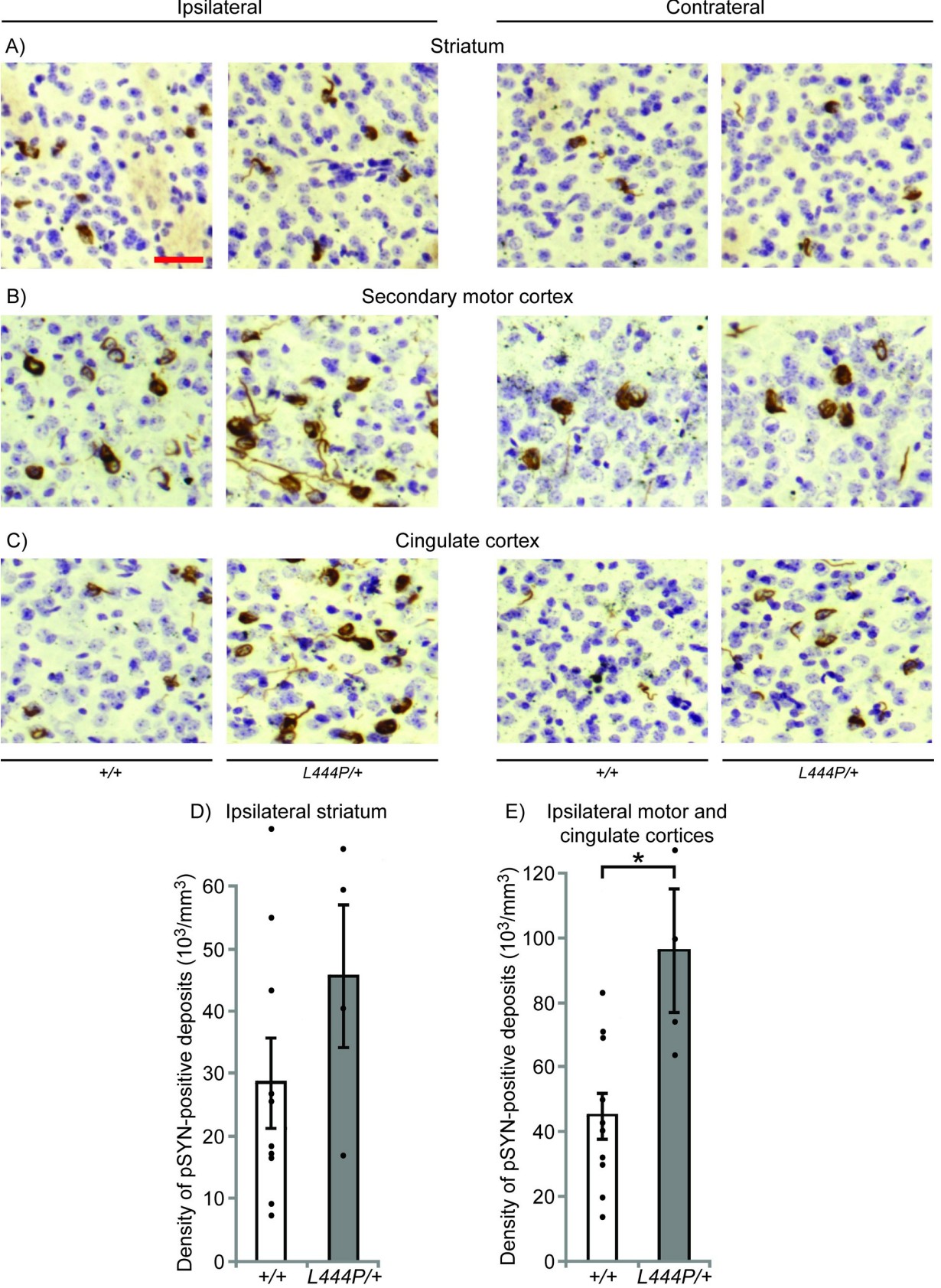

D) Ipsilateral striatum

E) Ipsilateral motor and cingulate cortices

**Fig 2. p-αSYN inclusions in the striatum and cortex (secondary motor and cingulate cortices) in αSYN-PFF-injected wild-type and *L444P/+* mice. (A)** Increased p-αSYN pathology in the striatal tissue at the level of 0.6mm anterior to the injection site (+0.8mm from the bregma) in the ipsilateral and contralateral hemispheres of *L444P/+* mice compared to their wild-type control littermates. **(B)** Increased p-αSYN pathology in the cortical tissue at the level of 0.6mm anterior to the injection site (+0.8mm from the bregma) in layer 5 of the secondary motor cortex in the ipsilateral and in layer 2 of the secondary motor cortex in the contralateral hemisphere of *L444P/+* mice compared to their wild-type control littermates. **(C)** Increased p-αSYN pathology in the cortical tissue at the level of 0.6mm anterior to the injection site (+0.8mm from the bregma) in the cingulate cortex in the ipsilateral and contralateral hemispheres of *L444P/+* mice compared to their wild-type control littermates. **(D)** Quantification of p-αSYN-positive deposits in the ipsilateral striatum of wild-type and *L444P/+* mice reveals a non-statistically significant difference between the genotypes (Student *t*-test, p = 0.23). **(E)** Quantification of p-αSYN-positive deposits in the ipsilateral motor and cingulate cortices of wild-type and *L444P/+* mice reveals a statistically significant difference between the genotypes (Student *t*-test, *p = 0.008). Scale bars = 25μm. Representative images shown. Ten +/+ and four *L444P/+* mice were analyzed in total.

significant increase in the number of p-αSYN deposits was observed in the cortex of *L444P/+* mice compared to wild-type littermates (45000±7000 vs. 96000±19000 LB-like inclusions/mm$^3$) (Student *t*-test, p = 0.008) (Fig 2E).

No p-αSYN pathology was observed after PBS or α-synuclein monomer injection in the brains of *L444P/+* and wild-type mice (data not shown).

## Discussion

This is the first study to analyze the effect of mouse αSYN-PFFs in the brains of *Gba1 L444P/+* mutant mice. Using this model, we showed that GCase deficiency greatly increases formation of pathological p-αSYN deposits in transgenic mice following αSYN-PFF injection.

We observed widespread p-αSYN pathology in the form of LB- and LN-like deposits throughout the brains of wild-type and *L444P/+* mice four months post-injection with αSYN-PFFs, but not with PBS or α-synuclein monomers. This observation is in line with previous studies, which showed that a single intracerebral injection of αSYN-PFFs is sufficient to induce α-synuclein spreading in wild-type mice, and to accelerate progression of PD-like pathology in transgenic mice expressing the human α-synuclein A53T mutation [4, 5, 8]. Our results further confirm that αSYN-PFFs are capable of inducing α-synuclein spreading and accumulation, leading to robust α-synuclein pathology in anatomically-interconnected brain regions.

We next assessed the effects of αSYN-PFFs on the formation of p-αSYN inclusions in the absence and presence of GCase deficiency. The aim of this was to determine whether the 30% decrease of GCase activity observed in the brains of *L444P/+* mice would enhance α-synuclein pathology *in vivo* [3]. We observed more prominent accumulation of p-αSYN deposits in several brain regions (including the motor, cingulate, perirhinal and entorhinal cortices, and the amygdala and substantia nigra pars compacta) in *L444P/+* mice compared to wild-type littermates. We then determined the number of LB- and LN-like deposits in the striatum and cortical area of *L444P/+* and wild-type mice, and observed significant increases in the number of p-αSYN deposits in the cortex of *L444P/+* mice. Taken together, these results indicate that GCase deficiency considerably increases accumulation and spread of pathological α-synuclein.

Several lines of evidence might explain how GCase deficiency increases formation of p-αSYN deposits. It has been shown that GCase reduction alters the formation and/or stability of α-synuclein polymers through glycosphingolipid accumulation, increasing the level of α-synuclein monomers that might subsequently misfold and aggregate into p-αSYN inclusions [9]. It has also been reported that in neuronal cultures derived from mice containing the *Gba1* L444P mutation and human α-synuclein, GCase deficiency significantly decreases the rate of α-synuclein degradation leading to α-synuclein accumulation [10]. Moreover, it has been suggested that the L444P mutation might increase total α-synuclein levels by prolonging the half-life of both endogenous α-synuclein and externally-delivered α-synuclein possibly by reducing

its lysosomal degradation [3]. In turn, these increased intraneuronal α-synuclein levels might promote α-synuclein assembly, subsequently enhancing α-synuclein aggregation into p-αSYN inclusions. These explanations may well also apply to αSYN-PFFs [11].

Altogether, our results indicate that GCase deficiency considerably enhances accumulation of pathological α-synuclein and favors spreading of its aggregates. It is of interest that the results described here have a clinical correlate in that PD patients with *L444P Gba1* mutations have earlier onset of disease, more rapid progression and increased cognitive dysfunction [12]. Our findings offer novel insight into how *Gba1* mutations might contribute to PD development and progression.

## Supporting information

**S1 Fig. p-αSYN inclusions in the perirhinal cortex, amygdala and substantia nigra in αSYN-PFF-injected wild-type and *L444P/+* mice.** (**A**) Increased p-αSYN pathology in the perirhinal cortex at the level of 2.2mm posterior to the injection site (-2.0mm from the bregma) in the ipsilateral hemisphere of *L444P/+* mice compared to their wild-type control littermates. (**B**) Increased p-αSYN pathology in the lateral amygdaloid nuclei at the level of 2.2mm posterior to the injection site (-2.0mm from the bregma) in the ipsilateral hemisphere of *L444P/+* mice compared to their wild-type control littermates. (**C**) Increased p-αSYN pathology in substantia nigra pars compacta at the level of 3.7mm posterior to the injection site (-3.5mm from the bregma) in the ipsilateral hemisphere of *L444P/+* mice compared to their wild-type control littermates. Scale bars = 100μm. Representative images shown. In total ten +/+ and four *L444P/+* mice were analyzed.
(TIF)

**S2 Fig. p-αSYN inclusions in the striatum and cortex (secondary motor and cingulate cortices) in αSYN-PFF-injected wild-type and *L444P/+* mice. (A)** Increased p-αSYN pathology in the striatal tissue at the level of 0.6mm anterior to the injection site (+0.8mm from the bregma) in the ipsilateral and contralateral hemispheres of *L444P/+* mice compared to their wild-type control littermates. (**B**) Increased p-αSYN pathology in the cortical tissue at the level of 0.6mm anterior to the injection site (+0.8mm from the bregma) in layer 5 of the secondary motor cortex in the ipsilateral and in layer 2 of the secondary motor cortex in the contralateral hemisphere of *L444P/+* mice compared to their wild-type control littermates. (**C**) Increased p-αSYN pathology in the cortical tissue at the level of 0.6mm anterior to the injection site (+0.8mm from the bregma) in the cingulate cortex in the ipsilateral and contralateral hemispheres of *L444P/+* mice compared to their wild-type control littermates. Scale bars = 100μm. Representative images shown. Ten +/+ and four *L444P/+* mice were analyzed in total.
(TIF)

**S3 Fig. Lack of p-αSYN inclusions in the perirhinal cortex, cingulate cortex and striatum in PBS-injected wild-type mice.** (**A**) No p-αSYN pathology in the perirhinal cortex at the level of 2.2mm posterior to the injection site (-2.0mm from the bregma) in the ipsilateral hemisphere of wild-type controls. (**B**) No p-αSYN pathology in the cortical tissue at the level of 0.6mm anterior to the injection site (+0.8mm from the bregma) in the cingulate cortex in the ipsilateral hemisphere of wild-type controls. (**C**) No p-αSYN pathology in the striatal tissue at the level of 0.6mm anterior to the injection site (+0.8mm from the bregma) in the ipsilateral hemispheres of wild-type controls. (**B**) Scale bars at low magnification = 100μm. Scale bars at high magnification = 25μm. Representative images shown. Ten wild-type mice were analyzed in total.
(TIF)

**S4 Fig. Additional images of p-αSYN inclusions in the striatum, amygdala and substantia nigra in αSYN-PFF-injected wild-type and *L444P*/+ mice.** (**A**) Increased p-αSYN pathology in the striatal tissue at the level of 0.6mm anterior to the injection site (+0.8mm from the bregma) in the ipsilateral and contralateral hemispheres of *L444P*/+ mice compared to their wild-type control littermates. (**B**) Increased p-αSYN pathology in the lateral amygdaloid nuclei at the level of 2.2mm posterior to the injection site (-2.0mm from the bregma) in the ipsilateral hemisphere of *L444P*/+ mice compared to their wild-type control littermates. (**C**) Increased p-αSYN pathology in substantia nigra pars compacta at the level of 3.7mm posterior to the injection site (-3.5mm from the bregma) in the ipsilateral hemisphere of *L444P*/+ mice compared to their wild-type control littermates. Scale bars = 50μm. Representative images shown. In total ten +/+ and four *L444P*/+ mice were analyzed.
(TIF)

**S5 Fig. Additional images of p-αSYN inclusions in the cortex (secondary motor and cingulate cortices) in αSYN-PFF-injected wild-type and *L444P*/+ mice.** (**A**) Increased p-αSYN pathology in the cortical tissue at the level of 0.6mm anterior to the injection site (+0.8mm from the bregma) in layer 5 of the secondary motor cortex in the ipsilateral and in layer 2 of the secondary motor cortex in the contralateral hemisphere of *L444P*/+ mice compared to their wild-type control littermates. (**B**) Increased p-αSYN pathology in the cortical tissue at the level of 0.6mm anterior to the injection site (+0.8mm from the bregma) in the cingulate cortex in the ipsilateral and contralateral hemispheres of *L444P*/+ mice compared to their wild-type control littermates. Scale bars = 50μm. Representative images shown. Ten +/+ and four *L444P*/+ mice were analyzed in total.
(TIF)

## Author Contributions

**Conceptualization:** Anna Migdalska-Richards, Anthony H. V. Schapira.

**Data curation:** Anna Migdalska-Richards, Michal Wegrzynowicz.

**Formal analysis:** Anna Migdalska-Richards, Michal Wegrzynowicz.

**Funding acquisition:** Anthony H. V. Schapira.

**Investigation:** Anna Migdalska-Richards, Michal Wegrzynowicz, Ian F. Harrison.

**Methodology:** Anna Migdalska-Richards.

**Resources:** Guglielmo Verona, Vittorio Bellotti, Anthony H. V. Schapira.

**Writing – original draft:** Anna Migdalska-Richards.

**Writing – review & editing:** Anna Migdalska-Richards, Michal Wegrzynowicz, Ian F. Harrison, Vittorio Bellotti, Maria Grazia Spillantini, Anthony H. V. Schapira.

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
