## [Decision Letter · Decision Letter 0]

4 Jun 2020

PONE-D-20-14240

L444P Gba1 mutation increases formation and spread of α-synuclein deposits in mice injected with mouse α-synuclein pre-formed fibrils

PLOS ONE

Dear Dr. Schapira,

Thank you for submitting your manuscript to PLOS ONE. After careful consideration, we feel that it has merit but does not fully meet PLOS ONE’s publication criteria as it currently stands. Therefore, we invite you to submit a revised version of the manuscript that addresses the points raised during the review process.

Below you will find comments from the Academic editor and an expert reviewer. The editor and reviewer are both concerned about the relatively small number of L444P/+ animals that were analyzed, and there are some questions about statistical methods or interpretation. The reviewer suggests validation of the histology data with a second antibody, which would be optional for the authors in accord with journal policy.

We look forward to receiving your revised manuscript.

Kind regards,

David R Borchelt

Academic Editor

PLOS ONE

Additional Editor Comments:

The manuscript is approaching the bare minimum of a publishable unit of information. Can the authors provide a justification for the low "n" number for L444P/+ mice that were injected an analyzed? Four is a relatively small number. I am not entirely convinced that the differences in pathology between genotypes is so robust than an n of 4 is really sufficient to draw hard conclusions. The data in Figure 2D and E should be presented as scatter plots so that the reader can visualize the variability in the data. On page 8, line 182 the authors state that "the number of p-a-Syn deposits was elevated in the striatum of L444P/+ mice compared to WT littermates, but this was not statistically significant". These two phrases are mutually exclusive. If the statistics do not confirm that the level of deposits was elevated, then one can not make the statement as written.

Because you have unequal sample sizes, it is necessary to do a post hoc test of validity. If the variances in the data sets are similar, then you can use Tukey-Kramer or Fisher's test among other possibilities. If the variances in the data are not equal, then there are other post hoc tests that must be used.

With the journals making it relatively painless to provide supplementary data, it would be useful to show some images of the absence of pathology in the controls as supplemental data. Similarly, with an N of only 4, more images of pathology could be provided as supplemental figures.

The resolution of the supplemental images is relatively low (even after downloading the better quality image).

Reviewers' comments:

Reviewer's Responses to Questions

**Comments to the Author**

1. Is the manuscript technically sound, and do the data support the conclusions?

Reviewer #1: Partly

2. Has the statistical analysis been performed appropriately and rigorously? 

Reviewer #1: Yes

3. Have the authors made all data underlying the findings in their manuscript fully available?

Reviewer #1: Yes

4. Is the manuscript presented in an intelligible fashion and written in standard English?

Reviewer #1: Yes

5. Review Comments to the Author

Reviewer #1: This is a manuscript by Migdalska-Richards and colleagues reporting that a single injection of pre-formed mouse α-synuclein fibrils into the striatum of GBA1 mice that carry a L444P knock-in mutation compared to wild-type littermates result in a greater formation and spread of α-synuclein inclusions. The following issues need to be addressed:

1) Why were only 4 L444P/+ mice used compared to 10 wild-type littermates?

2) Only one antibody to pSer129 was used for immunostaining. For proper scientific validation and rigor, the findings should be confirmed with at least one addition antibody, preferably not to pSer129.

3) In Figure 2E, the quantification of the motor and cingulate cortices were combined. These should be separate.

6. PLOS authors have the option to publish the peer review history of their article (what does this mean?). If published, this will include your full peer review and any attached files.

Reviewer #1: No

---

## [Author Response · Author response to Decision Letter 0]

27 Jul 2020

We thank both the Editor and Reviewer for their comments.

Answers to both the Editor and Reviewer:

1. Why the relatively small number of L444P/+ animals that were analysed?

We used a relatively small number of L444P/+ mice based on previous publications concerning pre-formed fibril injections and our own published data.

Luk, KC, Kehm V, Carroll J, Zhang B, O’Brien P, Trojanowski JQ, et al. Pathological α-synuclein transmission initiates Parkinson’s-like neurodegeneration in nontransgenic mice. Science. 2012a 338(6109):949-53.

In this paper, 3–7 animals were examined per group.

Luk, KC, Kehm VM, Zhang B, O'Brien P, Trojanowski JQ, Lee VM. Intracerebral inoculation of pathological α-synuclein initiates a rapidly progressive neurodegenerative α-synucleinopathy in mice. J Exp Med. 2012b 209(5):975-86.

In this paper, 3–12 animals were examined per group.

Migdalska-Richards AM, Wegrzynowicz M, Rusconi R, Deangeli G, Di Monte DA, Spillantini MG, et al. The L444P Gba1 mutation enhances α-synuclein induced loss of nigral dopaminergic neurons in mice. Brain. 2017 140:2706-2721.

In this paper, 3–6 animals were examined per group.

Based on the number of mice used in each of these 3 publications (please see above) and that with careful measurements, the number was (and is) expected to produce a statistically significant result. Also, as our funders (PUK) and institution (UCL) are committed to the 3Rs, the minimum number of animals was entered into the design.

Answers to the Editor:

1. The data in Figure 2D and E should be presented as scatter plots so that the reader can visualize the variability in the data. 

The figures were changed accordingly.

2. On page 8, line 182 the authors state that "the number of p-a-Syn deposits was elevated in the striatum of L444P/+ mice compared to WT littermates, but this was not statistically significant". These two phrases are mutually exclusive. If the statistics do not confirm that the level of deposits was elevated, then one cannot make the statement as written.

The statement was changed to: “There was a trend for an increase in p-αSYN deposits in the striatum of L444P/+ mice compared to WT littermates, but this was not statistically significant.”

3. Because you have unequal sample sizes, it is necessary to do a post hoc test of validity. If the variances in the data sets are similar, then you can use Tukey-Kramer or Fisher's test among other possibilities. If the variances in the data are not equal, then there are other post hoc tests that must be used.

Typically, a post hoc test of validity is applied when ANOVA test is used to calculate a statistical significance. However, post hoc tests are not commonly used when t-Test is used to calculate a statistical significance. t-Test is commonly used to compare statistical changes between two groups, while ANOVA is typically to compare statistical changes among more than two groups. As we only had two groups, we chose t-Test for our analysis. We checked the equality of variances using Levene's test and we did not find any statistically significant changes between any analysed groups (p=0.304 for cortex and p=0.884 for striatum). We checked the data normality using Shapiro-Wilk test and found the normal distribution in all analysed groups (p=0.579 for WT/cortex, p=0.716 for L444P/cortex, p=0.052 for WT/striatum and p=0.959 for L444P/striatum). Thus, after confirming the normal distribution and the equality of variances, we could compare all the groups.

4. With the journals making it relatively painless to provide supplementary data, it would be useful to show some images of the absence of pathology in the controls as supplemental data. Similarly, with an N of only 4, more images of pathology could be provided as supplemental figures.

The Supplementary Figure 3 was generated to show the absence of p-αSYN pathology in PBS-injected control mice. The Supplementary Figure 4 and 5 were generated to provide additional images showing the presence of p-αSYN pathology in αSYN-PFF-injected L444P/+ mice.

5. The resolution of the supplemental images is relatively low (even after downloading the better quality image).

The better-quality images were generated.

Answers to the Reviewer:

1. Only one antibody to pSer129 was used for immunostaining. For proper scientific validation and rigor, the findings should be confirmed with at least one addition antibody, preferably not to pSer129.

We used a well validated antibody for pSer129 (see publications above), and we are unclear as to why another antibody not to pSer129 would have been of value.

2. In Figure 2E, the quantification of the motor and cingulate cortices were combined. These should be separate.

 In the mouse, the cingulate and motor cortices are adjacent and the border between them often indistinct, making separate counting inappropriate. Therefore, for the purpose of illustrating pathology (see Figure 2) we selected representative sections taken from the periphery of the respective areas away from their border. However, for the purpose of quantitation we combined the two adjacent areas of cingulate and motor cortices to avoid mis-appropriation of data between the two.

---

## [Decision Letter · Decision Letter 1]

10 Aug 2020

L444P Gba1 mutation increases formation and spread of α-synuclein deposits in mice injected with mouse α-synuclein pre-formed fibrils

PONE-D-20-14240R1

Dear Dr. Schapira,

We’re pleased to inform you that your manuscript has been judged scientifically suitable for publication and will be formally accepted for publication once it meets all outstanding technical requirements.

Kind regards,

David R Borchelt

Academic Editor

PLOS ONE

Additional Editor Comments (optional):

Although Reviewer 1 remains dissatisfied, the authors have addressed all of the technical comments and provide an adequate justification for the animal numbers. Personally, I would still like to have seen a larger N for the transgenics, but I can't argue with the justification.

Reviewers' comments:

Reviewer's Responses to Questions

**Comments to the Author**

1. If the authors have adequately addressed your comments raised in a previous round of review and you feel that this manuscript is now acceptable for publication, you may indicate that here to bypass the “Comments to the Author” section, enter your conflict of interest statement in the “Confidential to Editor” section, and submit your "Accept" recommendation.

Reviewer #1: (No Response)

2. Is the manuscript technically sound, and do the data support the conclusions?

Reviewer #1: Partly

3. Has the statistical analysis been performed appropriately and rigorously? 

Reviewer #1: Yes

4. Have the authors made all data underlying the findings in their manuscript fully available?

Reviewer #1: Yes

5. Is the manuscript presented in an intelligible fashion and written in standard English?

Reviewer #1: Yes

6. Review Comments to the Author

Reviewer #1: This is a resubmitted manuscript by Migdalska-Richards and colleagues. Some simple but scientifically important changes were requested that have not been addressed.

1) It was requested that for proper scientific validation and rigor, the findings be confirmed with at least one additional antibody that would preferably be against a difference epitope thus not another pSer129 specific antibody. Using more than one antibody for immunocytochemical validation is a standard pathological practice. The authors simply refused this request.

2) It was requested that the quantification of the motor and cingulate cortices be separated as these are distinct neuroanatomical structures with distinct functions. The authors refused to do this simple request and their explanation is that it is not appropriate as they claim that there can distinguish between these regions when taking images but that they could not do the same for quantification.

3) It is still not explained why N=10 wild-type littermates and only N =4 L444P/+ mice was used. Why the imbalance in the number of mice used?

7. PLOS authors have the option to publish the peer review history of their article (what does this mean?). If published, this will include your full peer review and any attached files.

Reviewer #1: No

---

## [Editor Report · Acceptance letter]

13 Aug 2020

PONE-D-20-14240R1 

L444P *Gba1* mutation increases formation and spread of α-synuclein deposits in mice injected with mouse α-synuclein pre-formed fibrils 

Dear Dr. Schapira:

I'm pleased to inform you that your manuscript has been deemed suitable for publication in PLOS ONE. Congratulations! Your manuscript is now with our production department. 

Kind regards, 

on behalf of

Prof. David R Borchelt 

Academic Editor

PLOS ONE